# Benchmarking Visual Knowledge in Multimodal Large Language Models

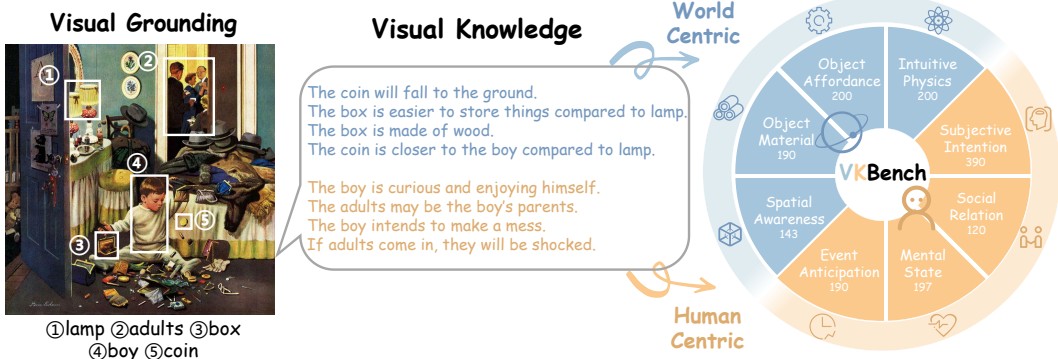

Figure 1: VKBench systematically evaluates visual knowledge across world-centric and human-centric tasks, evaluating MLLMs beyond simple perception. It marks a shift from seeing and recognizing to understanding the world principles, enabling models to achieve true visual comprehension.

## Abstract

While Multimodal Large Language Models (MLLMs) have become adept at recognizing objects, they often lack the intuitive, human-like understanding of the world's underlying physical and social principles. This capability, which we term *visual knowledge*, forms a bridge between perception and reasoning, yet remains an underexplored gap in current systems. To systematically measure this capability, we present **VKBench**, a comprehensive video benchmark featuring 1,680 questions in 1,249 videos, covering eight core types of visual knowledge spanning both *world-centric* (e.g., intuitive physics) and *human-centric* (e.g., subjective intentions). Results show that leading models still fall short of human performance, with particularly notable gaps in world-centric visual knowledge. To bridge this gap, we introduce **VKQA**, a new dataset, and **Video-VK+**, a baseline model that explicitly incorporates visual knowledge into MLLMs. Video-VK+ follows a structured *See–Think–Answer* format and adopts reinforcement learning with visual knowledge reward. This approach improves performance on VKBench by 3.7% and surpasses existing models on multiple video benchmarks. Our findings highlight visual knowledge as a key component for developing more robust and generalizable MLLMs that can not only see but also truly understand our world.

## 1 Introduction

Humans possess an intuitive understanding of the world, effortlessly predicting the trajectory of a bouncing ball or inferring the fragility of a glass from a single glance. This ability, often referred to as visual knowledge (Zhu et al., 2020; Wang et al., 2025d), represents an intermediate cognitive layer that internalizes the principles that govern both the physical and social worlds. For Multimodal Large Language Models (MLLMs), which frame predictions as probabilities conditioned on both visual evidence and world knowledge, this layer is crucial. By grounding reasoning in rich, visually-derived context, visual knowledge helps reduce over-reliance on brittle language priors and mitigates model hallucinations. Enhancing this capability is fundamental to advancing MLLMs' performance

on complex tasks such as compositional problem-solving (Yue et al., 2023; Zhao et al., 2025) and multi-hop reasoning (Cheng et al., 2025), towards human-like AI (Lee et al., 2024; Liu et al., 2025a).

Despite its importance, this critical dimension of visual understanding remains largely unexplored in the context of MLLMs. Existing research has primarily focused on enhancing fine-grained perception through techniques such as leveraging multi-scale features (Jiang et al., 2023b; Liu et al., 2024a), aligning vision and text with detailed descriptions (Zhang et al., 2024a; Wang et al., 2024a), or processing high-resolution inputs (Liu et al., 2024b; Wang et al., 2025a). While these approaches improve an MLLM's ability to recognize concrete objects and attributes, they contribute little to instilling the embedded, intuitive knowledge necessary for deeper reasoning.

In this paper, we first introduce VKBench to quantitatively measure this gap. VKBench is a comprehensive video benchmark designed to evaluate the visual knowledge of MLLMs. Spanning both *world-centric* (Intuitive Physics, Object Affordances, Object Materials, Spatial Awareness) and *human-centric* (Event Anticipation, Mental States, Social Relations, Subjective Intention) domains, VKBench comprises 1249 videos and 1680 multiple-choice questions across eight dimensions. Through rigorous data curation, we isolate visual knowledge from confounding cues like audio or language biases, creating a clean and challenging testbed. Our evaluation shows that even the most advanced MLLMs fall short of human performance with an overall gap of 15.0%, which is particularly pronounced in *world-centric* tasks: on Intuitive Physics and Spatial Awareness, model barely exceeds random guessing, trailing human performance by **38.5%** and **32.7%**, respectively. These findings highlight the limitation in current MLLMs' ability to comprehend visual knowledge.

Having quantified the problem, we then demonstrate that visual knowledge is learnable. We propose Video-VK+, a baseline model designed to explicitly integrate visual knowledge. Video-VK+ employs a structured *See–Think–Answer* reasoning format and is trained with reinforcement learning guided by a dedicated visual knowledge reward signal. To facilitate this, we introduce VKQA-30K, a new large-scale video corpus containing diverse instances of visual knowledge. When trained on VKQA-30K, Video-VK+ achieves a significant **3.7%** improvement on VKBench and shows strong generalization to other leading video understanding benchmarks, including MVBench (**+5.4%**) (Li et al., 2024c), Video-MME (**+7.0%**) (Fu et al., 2025) and MMVU (**+5.7%**) Zhao et al. (2025). These results underscore the pivotal role of explicit visual knowledge in advancing multimodal learning.

In summary, our contributions are as follows: (1) We formalize the concept of **visual knowledge** for MLLMs, highlighting the critical gap between perceptual accuracy and cognitive reasoning in current models. (2) We introduce **VKBench**, a comprehensive video benchmark covering eight dimensions of visual knowledge, carefully curated to minimize confounding cues and provide a challenging testbed. (3) We release **VKQA**, a large-scale video corpus rich in visual knowledge, and propose **Video-VK+**, a baseline model that explicitly integrates visual knowledge using a structured *See–Think–Answer* format and reinforcement learning with a visual knowledge reward.

## 2 RELATED WORK

**Multimodal Large Language Models.** The rapid rise of Large Language Models (LLMs) has spurred the development of Multimodal LLMs (MLLMs), which integrate visual perception with language reasoning to bridge cross-modal semantic gaps. Modern MLLMs (Liu et al., 2023; Dai et al., 2023; Ye et al., 2023; Wang et al., 2024b; Zeng et al., 2024; Bai et al., 2025; Zhu et al., 2025) typically employ lightweight alignment modules that map visual features from encoders like CLIP (Radford et al., 2021) or SigLIP (Zhai et al., 2023) into the embedding space of open-source LLMs (Jiang et al., 2023a; Dubey et al., 2024). Fine-tuned on image-text pairs, these models rival or surpass proprietary systems (Achiam et al., 2023; Team et al., 2024) on multimodal benchmarks. Recently, inspired by DeepSeek-R1 (Guo et al., 2025), RL post-training has been adapted to MLLMs via advanced reward design for vision reasoning (Huang et al., 2025; Yang et al., 2025c; Xia et al., 2025; Li et al., 2025b), and extended to video domains (Feng et al., 2025; Li et al., 2025a; Wang et al., 2025b), marking a shift toward structured reasoning in dynamic multimodal contexts.

**Benchmarks for Multimodal Large Language Models.** MLLM evaluation has rapidly evolved alongside model capabilities. Early benchmarks focused on visual perception, such as VQA (Antol et al., 2015), captioning (Lin et al., 2014), and OCR (Singh et al., 2019). Later works like Fu et al. (2023); Liu et al. (2023); Li et al. (2024b); Fu et al. (2024); Song et al. (2024) decoupled perception

from reasoning in static images, while POPE (Li et al., 2023; Guan et al., 2024) exposed hallucinations via adversarial questioning. In video, the evaluation shifts to temporal reasoning, advancing from single-scene tasks (Goyal et al., 2017; Xiao et al., 2021) to complex, long-form understanding (Li et al., 2024c; Liu et al., 2024c). Recent benchmarks emphasize multimodal reasoning (Yu et al., 2025; Liu et al., 2025b) and knowledge-intensive tasks (Zhang et al., 2024b; Zhao et al., 2025), integrating visual evidence with expert-level knowledge to test real-world applicability. In contrast, the questions in VKBench focus on visual knowledge rooted in human psychology and are deliberately formulated to be straightforward, eliminating the need for domain-specific knowledge.

**Visual Knowledge.** David Marr's classical formulation of computer vision (Marr, 2010) focuses on identifying *what* and *where*, but human visual intelligence encompasses much more. Zhu et al. (2020) proposed FPICU as the *dark matter* of vision, like dark matter in the universe, invisible in pixels, yet essential for meaningful understanding. This insight has also been introduced by the concept of visual knowledge (Pan, 2019; Wang et al., 2025d), which not only emerges from perception but also enables visual memory, and mental simulation, forming the basis of how we interpret the world. VCR (Li et al., 2022) introduced the visual commonsense reasoning task, yet its scope is limited to scenarios involving human activities. Recently, Li et al. (2024d) uncovered the core knowledge deficits in MLLMs where they consistently underperform on low-level abilities relative to high-level ones, highlighting a persistent disconnect between seeing and truly understanding.

## 3 VKBENCH

We introduce VKBench, a comprehensive video benchmark designed to evaluate the visual knowledge of MLLMs across eight dimensions. In this section, we first formally describe the semantic scope of visual knowledge, followed by an overview of each task included in VKBench. We then provide a detailed account of the dataset construction and the QA selection pipeline. Finally, we evaluate a range of state-of-the-art MLLMs on VKBench and analyze their performance.

### 3.1 VISUAL KNOWLEDGE

Visual knowledge extends far beyond the mere perception or grounding of concrete objects, encompassing a set of abstract and transferable principles rooted in cognitive psychology. These principles shape how we interpret our surroundings, engage in reasoning, and act upon the world. Generally, humans intuitively grasp the underlying patterns of real-world phenomena as self-evident truths, whereas machines cannot directly uncover them merely by recognizing pixels or physical entities. For example, when a moving sphere rises instead of falling under gravity, we immediately recognize it as a violation of physical laws; when seeing transparent, colorless crystals on a table, we readily identify them as glass and infer their fragility relative to metal; and when observing someone approach a door, we naturally infer their intention to open it and leave. While these judgments are grounded in observable, pixel-level entities, they rely even more heavily on the invisible scaffolding of visual knowledge. When encountering such scenarios, humans often arrive at an immediate and intuitive understanding, akin to a form of visual commonsense (Zellers et al., 2019) or a visual conditioned reflex (Salter, 2001), without requiring any expert-level domain knowledge.

### 3.2 TASK DEFINITION

Based on the meaning of visual knowledge, we concretize its domain by giving its eight most related tasks in VKBench, involving both physical reality (***world-centric***) and psychologically grounded human intentions (***human-centric***). The overview of VKBench is shown in Figure 2.

**World-Centric.** The *world-centric* categories assess MLLMs' capacity to comprehend the objective physical structure and law of the environment. They are grounded in physics, material properties, and spatial relationships, and are largely independent of human psychological or cultural context. Specifically, they include four core competencies: (1) **Intuitive Physics**: Judge the physical plausibility of dynamic events, following principles such as object permanence, immutability, continuity, gravity and so on. (2) **Object Affordance**: Determine potential functional uses based on objects' perceptual and structural properties. (3) **Object Material**: Identify the material composition of objects by leveraging either observable intrinsic properties or behavioral cues revealed

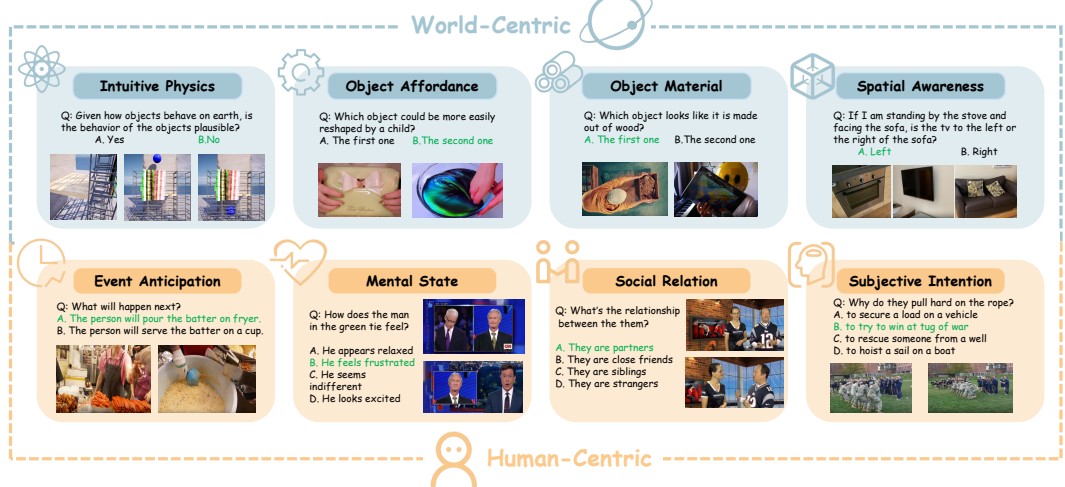

Figure 2: An overview of the VKBench. Video and QA case of each tasks are illustrated.

through human-object interactions. (4) **Spatial Awareness**: Infer relative spatial relationships, such as positions, directions, and navigation within an environment.

**Human-Centric.** The *human-centric* categories evaluate human agents, inferred from visuals such as facial expressions, body postures, gaze, motion trajectories, and social configurations. Contrary to physical knowledge, this is inherently interpretive, culturally nuanced, and psychologically layered. It demands that models reconstruct the invisible mental and social states from visible actions or activities. Core competencies include: (5) **Event Anticipation**: Infer the most probable subsequent events based on existing video clues and social norms. (6) **Mental State**: Infer internal psychological states of individuals, as well as the affective atmosphere of the surrounding environment. (7) **Social Relation**: Infer interpersonal relationships and social roles in human society. (8) **Subjective Intention**: Reconstruct the underlying goals or motivations behind human observed actions.

### 3.3 VKBench Construction

**Data Collection.** To construct a high-quality and unambiguous benchmark that comprehensively evaluates diverse dimensions of visual knowledge, we carefully curate data from multiple established datasets aligned with the task definition of VKBench. Specifically, we draw on videos and associated annotations from IntPhys 2 (Bordes et al., 2025), PACS (Yu et al., 2022), VSI-Bench (Yang et al., 2025b), VLEP (Lei et al., 2020), Social-IQ 2.0 (Wilf et al., 2023), and RexTime (Chen et al., 2024). Each sample in VKBench is structured as a multiple-choice question designed to probe a specific facet of visual knowledge.

**Filtering.** The filtering principle is to directly evaluate how MLLMs work on visual knowledge, minimizing the impact brought by text priors or reasoning effects from LLMs. Usually, vqa related to *world-centric* types contain no context in the question or answer candidates, thus they are less concerned in this process. In contrast, *human-centric* categories are often heavily biased: **(1)** A number of questions inherently require complementary audio cues more than pure visuals, introducing extra reliance. **(2)** Such questions frequently embed implicit information where MLLMs may exploit language shortcuts, leveraging world knowledge of LLM for response. To alleviate the above problems, we design a progressive filtering pipeline, leveraging multiple MLLMs and LLMs to reduce unwanted biases step by step, illustrated in Figure 3 and detailed as follows.

- **I. Minimize Audio Reliance.** We abandon QAs whose answers are relevant to their corresponding audios to some degree. First, we transcribe the audio track of each video using Whisper-large-v2 (Radford et al., 2022) to generate subtitles. We then compute the semantic similarity between these subtitles and the ground-truth answers using Qwen3-8B-Embedding (Zhang et al., 2025). Any question whose corresponding subtitle yields a similarity score exceeding 0.3 is discarded, as such questions may be answerable through audio cues alone.

- **II. Reduce Language Bias.** We drop QAs whose questions are easily solved by only answer candidates without visuals, inspired by MMMU-Pro (Yue et al., 2024). Specifically, we prompt

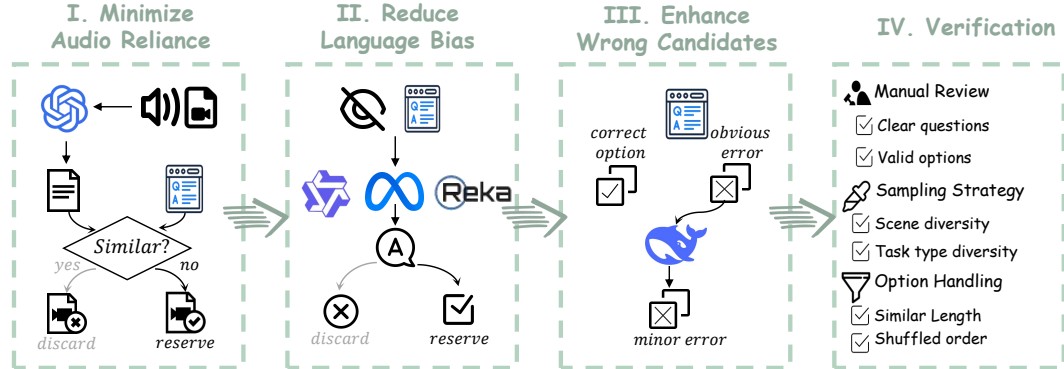

Figure 3: An overview of the QA filtering pipeline.

three powerful open-source LLMs, Qwen3-32B-thinking (Yang et al., 2025a), Llama-3.3-70B-Instruct (Grattafiori et al., 2024), and Reka-Flash-3 (RekaAI, 2025), to answer each question in the absence of any visual input. Models are instructed to provide responses even when they explicitly acknowledged the need for visual context, simulating a "blind VQA" scenario. Each model generated ten independent answers per question; if a model produced the correct answer more than five times, the question was deemed linguistically answerable. Questions flagged as answerable by at least two out of the three models were excluded from the final benchmark.

- **III. Enhance Wrong Candidates.** For the remaining QAs from the previous two steps, we leverage DeepSeek-R1 (Guo et al., 2025) to refine the incorrect options while preserving the original correct answers. These distractors are crafted to be semantically plausible yet subtly incorrect, ensuring sufficient choice discrimination to elevate the challenge and better test visual knowledge.

- **IV. Human Verification.** Following the shuffling of multiple-choice options to eliminate positional bias, we conduct a comprehensive human review to validate the quality and fairness of all questions. This process culminated in the construction of VKBench, a benchmark comprising 1249 video clips and 1680 rigorously curated question-answer pairs spanning eight distinct dimensions of visual knowledge. More details of VKBench are in Appendix E.

## 3.4 EVALUATION

We benchmark a diverse set of multimodal foundation models capable of processing video or multi-image inputs on VKBench. We evaluate 23 models covering both open-source and advanced proprietary MLLMs. On the open-source side, the evaluated models include VideoLLaMA2 (Cheng et al., 2024), mPLUG-Owl3-7B (Ye et al., 2024), MiniCPM-V-2.6 & 4.5 (Yao et al., 2024), LLaVA-OV (Li et al., 2024a), LLaVA-Video (Zhang et al., 2024c), Qwen2.5-VL (Bai et al., 2025), InternVL-3.5 (Wang et al., 2025c), GLM-4.1V-9B-Thinking (Hong et al., 2025) and MiMo-VL-RL (Team et al., 2025). For advanced proprietary models, we include GPT-4o (Hurst et al., 2024), Gemini-2.5-Flash and Pro (Comanici et al., 2025). To ensure comparability, we restrict the maximum number of video input frames to 32. Further details of the implementation are provided in Appendix G.2.

**Overall Performance.** The evaluation results on VKBench are summarized in Table 1. Most models achieve overall accuracy between 60% and 70%. The strongest model, *InternVL3.5-241B-A28B*, reaches 74.6% but still trails human performance by **15.0%**. A notable performance disparity emerges when we categorize the tasks. On *human-centric* tasks, models demonstrate strong capabilities, with the leading model's accuracy of 81.9% closely approaching the human benchmark of 86.8%. We hypothesize this strength stems from the abundance of human social interactions and societal norms in the training data. In sharp contrast, models struggle significantly with *world-centric* tasks. The highest accuracies in the IP (59.0%) and SA (62.7%) domains fall short of human performance by substantial margins of **38.5%** and **32.7%**, respectively. This gap highlights a fundamental limitation in current MLLMs regarding the acquisition of knowledge about the physical world. We posit that this deficiency arises because much *world-centric* knowledge is grounded in physical perception, which is difficult to capture exhaustively through textual descriptions alone. The results

Table 1: Evaluation results on VKBench. Abbreviations adopted: IP for Intuitive Physics; OA for Object Affordance; OM for Object Material; SA for Spatial Awareness; EA for Event Anticipation; MS for Mental State; SR for Social Relation; SI for Subjective Intention; WC for overall accuracy on World-Centric; HC for overall accuracy on Human-Centric; Human Performance are sourced from original annotation or researchers' responses. Green marks the best results.

| Models | Overall | IP | OA | OM | SA | WC | EA | MS | SR | SI | HC |
|---|---|---|---|---|---|---|---|---|---|---|---|
| Random Guess | 36.7 | 50.0 | 50.0 | 50.0 | 31.9 | 45.5 | 50.0 | 25.0 | 25.0 | 25.0 | 30.3 |
| Human Performance | 89.6 | 97.5 | 90.5 | 96.8 | 95.4 | 95.0 | 81.1 | 83.2 | 86.7 | 87.2 | 86.8 |
| *Open-Source MLLMs* | | | | | | | | | | | |
| VideoLLaMA2-7B | 55.7 | 46.0 | 53.0 | 51.1 | 40.9 | 47.8 | 68.4 | 64.5 | 57.5 | 60.5 | 63.5 |
| MiniCPM-V 2.6 | 63.8 | 53.0 | 51.5 | 76.5 | 36.5 | 54.3 | 74.0 | 66.5 | 75.0 | 66.5 | 69.2 |
| MiniCPM-V 4.5 | 67.6 | 50.0 | 58.5 | 83.5 | 39.0 | 57.6 | 78.5 | 77.2 | 76.7 | 72.5 | 75.4 |
| mPLUG-Owl3-7B | 64.2 | 57.0 | 55.0 | 72.6 | 36.3 | 55.2 | 77.9 | 73.1 | 79.2 | 66.7 | 72.1 |
| LLaVA-OV-7B | 64.9 | 52.5 | 56.0 | 83.0 | 38.0 | 57.2 | 73.0 | 72.1 | 75.8 | 62.5 | 68.6 |
| LLaVA-OV-72B | 68.0 | 56.0 | 60.0 | 83.0 | 41.0 | 59.9 | 76.0 | 71.6 | 83.3 | 69.0 | 73.0 |
| LLaVA-Video-7B | 66.0 | 50.5 | 56.5 | 79.0 | 42.0 | 56.9 | 72.5 | 69.5 | 80.0 | 69.5 | 71.5 |
| LLaVA-Video-72B | 70.5 | 57.0 | 65.5 | 86.0 | 44.5 | 63.1 | 78.0 | 72.6 | 84.2 | 73.8 | 75.8 |
| Qwen2.5-VL-3B-Instruct | 61.1 | 50.0 | 57.0 | 77.9 | 38.3 | 55.7 | 71.6 | 73.1 | 71.7 | 57.4 | 65.8 |
| Qwen2.5-VL-7B-Instruct | 64.0 | 48.0 | 53.0 | 82.6 | 35.2 | 54.5 | 79.5 | 75.1 | 77.5 | 65.6 | 72.2 |
| Qwen2.5-VL-32B-Instruct | 64.5 | 53.0 | 57.5 | 76.8 | 39.4 | 56.6 | 77.4 | 76.7 | 75.8 | 64.4 | 71.4 |
| Qwen2.5-VL-72B-Instruct | 70.1 | 59.0 | 64.5 | 86.3 | 40.9 | 62.6 | 76.8 | 79.7 | 82.5 | 73.3 | 76.7 |
| MiMo-VL-7B-RL | 68.0 | 56.0 | 66.0 | 80.5 | 49.2 | 62.8 | 78.4 | 75.6 | 80.8 | 65.4 | 72.5 |
| GLM-4.1V-9B-Thinking | 68.3 | 54.0 | 66.5 | 80.5 | 47.0 | 61.9 | 72.5 | 73.1 | 80.8 | 69.8 | 72.6 |
| InternVL3.5-8B | 66.9 | 50.5 | 58.0 | 78.9 | 44.6 | 57.9 | 75.3 | 75.6 | 75.8 | 73.8 | 74.8 |
| InternVL3.5-8B-Think | 67.6 | 50.5 | 63.0 | 80.5 | 42.0 | 58.9 | 76.3 | 74.1 | 78.3 | 74.1 | 75.1 |
| InternVL3.5-30B-A3B | 71.5 | 49.5 | 65.0 | 82.1 | 62.7 | 64.6 | 75.8 | 77.2 | 81.7 | 77.2 | 77.5 |
| InternVL3.5-38B | 72.7 | 49.0 | 61.0 | 82.6 | 59.1 | 62.7 | 77.9 | 76.1 | 82.5 | 85.4 | 81.4 |
| InternVL3.5-38B-Think | 71.8 | 49.0 | 66.5 | 79.0 | 54.9 | 62.2 | 80.5 | 76.1 | 80.8 | 82.1 | 79.7 |
| InternVL3.5-241B-A28B | 74.6 | 52.5 | 67.5 | 85.8 | 60.6 | 66.4 | 81.6 | 77.2 | 84.2 | 83.6 | 81.9 |
| *Proprietary MLLMs* | | | | | | | | | | | |
| GPT-4o | 65.7 | 55.0 | 74.0 | 84.7 | 42.5 | 64.0 | 74.7 | 63.5 | 65.0 | 65.9 | 67.1 |
| Gemini-2.5-Flash | 68.8 | 56.0 | 80.0 | 90.0 | 51.8 | 69.3 | 76.8 | 65.5 | 74.2 | 63.6 | 68.2 |
| Gemini-2.5-Pro | 71.1 | 55.0 | 82.0 | 88.4 | 55.4 | 70.1 | 71.6 | 73.1 | 75.8 | 70.3 | 72.0 |

suggest that the standard next-word prediction training objective may not provide sufficient supervisory signals for learning these concepts. This indicates that incorporating world models could be promising for these limitations.

**Proprietary vs Open-Source MLLMs.** Proprietary models demonstrate clear advantages on *world-centric* tasks, such as Object Affordance, Object Material, and Spatial Awareness, likely benefiting from richer pre-training corpora and proprietary alignment techniques optimized in physical world. However, these same models often underperform in *human-centric* visual knowledge, where open-source counterparts like *Qwen2.5-VL* and *InternVL3.5* excel. This asymmetry suggests divergent optimization goals: proprietary pipelines emphasize physics-based property, while open-source efforts prioritize socially diverse supervision. Bridging this gap will be critical for achieving balanced competence across the full spectrum of visual knowledge.

**LLM Scaling.** Scaling clearly improves performance. Larger models like *LLaVA-Video-72B* and *Qwen2.5-VL-72B-Instruct* consistently outperform smaller variants across most dimensions. This suggests that scaling the language component enables MLLMs to achieve a deeper understanding of visual knowledge. Joint training with a stronger LLM during large-scale alignment yields a more powerful vision encoder, further enhancing the model's ability to interpret visual knowledge.

**Thinking or Not.** *MiMo-VL-7B-RL*, *GLM-4.1V-9B-Thinking* and *InternVL3.5-8B-Think* exhibit a modest gain over their non-thinking counterparts, especially on *world-centric* tasks. However, *InternVL3.5-38B-Think* declines due to severe repetition. As VKBench involves relatively straightforward visual knowledge, excessive reasoning can be counterproductive, highlighting the need for an appropriate level of language reasoning to improve the understanding of visual knowledge.

**Correlation between Different Tasks.** As shown in Figure 5, the Pearson correlation coefficients among the eight tasks reveal two clusters, each exhibiting strong intra-cluster but weak inter-cluster

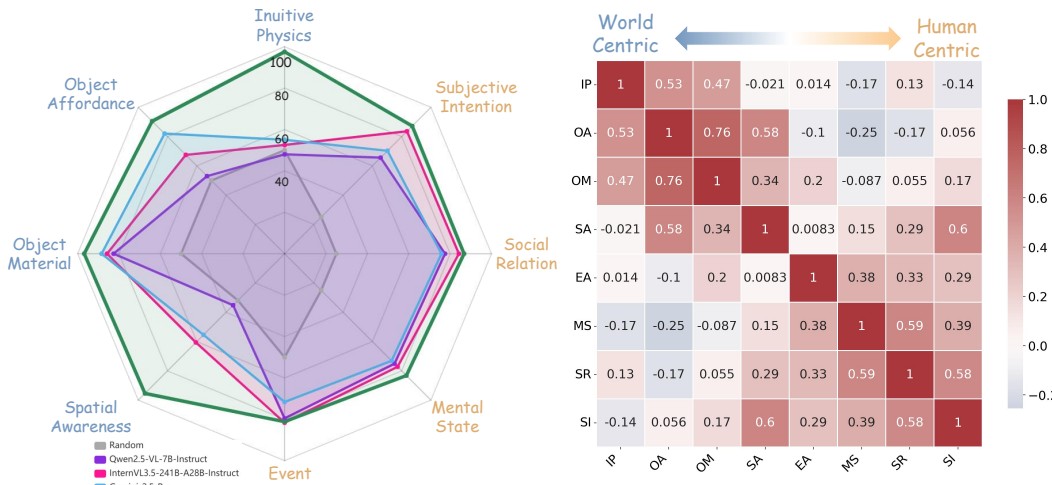

Figure 4: Accuracies of MLLMs on VKBench.

Figure 5: Pearson Correlation between 8 Visual Knowledge Tasks.

correlations. This clustering pattern corroborates VKBench's categorization of tasks into world-centric and human-centric domains, indicating these two dimensions of visual knowledge function as largely independent subsystems. See Appendix H for more analysis on VKBench.

## 4 EMBEDDING VISUAL KNOWLEDGE INTO MLLMS

Here we make initial attempts to explicitly integrate visual knowledge into MLLMs. Our core idea is to encourage MLLMs to rely more heavily on evidence rich in visual knowledge during the generation process, so that we can effectively offload the burden from the language prior, reducing hallucinations (see Appendix C for theoretical analysis). To operationalize this principle, we design an explicit reasoning pathway that anchors responses directly in perceived visual evidence, which is implemented through two key components: (1) a structured *See–Think–Answer* output format that enforces perceiving visual knowledge before reasoning, and (2) a visual knowledge reward based on the GRPO paradigm (see Appendix D). Together, these form the baseline we term Video-VK+.

***See-Think-Answer* Output Format.** Inspired by Xiao et al. (2025); Xia et al. (2025), we impose a constraint on Video-VK+ to explicitly generate a self-contained visual description prior to performing reasoning. This description content is designed to encapsulate the rich visual knowledge essential for the task, thereby laying a solid foundation for subsequent reasoning. To enforce strict adherence to the structured *See–Think–Answer* output, format reward $r_f$ is computed via regular expression matching over the model's output. For multiple-choice QA, we define an accuracy reward $r_a$ that equals 1 if the model's prediction matches the ground-truth answer, and 0 otherwise.

**Visual Knowledge Reward.** We employ an external frozen MLLM as the verifier model to assess whether the correct answer can be directly inferred from the generated visual description (Figure 12). To encourage the model to produce self-contained content rich in visual knowledge, we incorporate a binary visual knowledge reward $r_v \in \{0, 1\}$ into the policy update process, where $r_v = 1$ indicates that the visual content is sufficient for deriving the answer without further reasoning and $r_v = 0$ otherwise. This reward incentivizes the model to focus more on enriching its visual knowledge, the richer the visual knowledge, the less reasoning effort is required from the LLM. The final reward is formulated as:

$$R_i = r_f + r_a + \lambda \cdot r_v, \tag{1}$$

where $\lambda$ is a hyperparameter controlling the relative weight of the visual knowledge reward.

### 4.1 VKQA: A DATASET CENTERED ON VISUAL KNOWLEDGE

To meet the training requirements of Video-VK+, we construct a training dataset, named VKQA-30K, by curating and augmenting samples from open-source VLM benchmarks. Each instance in VKQA-30K consists of a question accompanied by multiple-choice options. We carefully perform

deduplication to ensure no overlap between VKQA-30K and VKBench, thereby avoiding data leakage and guaranteeing an **out-of-domain** evaluation setting. See Appendix F for details of VKQA.

**RL Data Collection.** Our reinforcement learning dataset comprises approximately 30K samples curated from publicly available VLM datasets, named VKQA-30K. The dataset covers three major domains: general (40%), world-centric (30%), and human-centric (30%). These instances encapsulate diverse and rich visual knowledge across the aforementioned dimensions.

**Cold Start Data Generation.** We also provide data in the *See–Think–Answer* format to facilitate cold-start scenarios. For each QA instance in VKQA-30K, we first employed an MLLM to generate *See-Think-Answer* responses and retained only those cases where the model produced both a correct answer and an output in desired format. Following the similar manner in Sec. 4, we then ensured the model could correctly answer question using only the text-based visual content as a proxy for the visual input, thereby guaranteeing that the reasoning was grounded in explicit visual knowledge. This process resulted in approximately 12K high-quality QAs, which we denote as VKQA-CS-12K.

## 5 EXPERIMENT

### 5.1 MAIN RESULTS

We adopt Qwen2.5-VL-7B-Instruct (Bai et al., 2025) as the base model for validation. To align the model with the structured *See–Think–Answer* output format, we first conduct a SFT cold-start phase. This is followed by training with GRPO augmented with the visual knowledge reward $r_v$, to encourage the model to generate responses by explicitly leveraging the visual knowledge it has acquired. This two-stage process yields the final Video-VK+.

Table 2: Results on VKBench and other video benchmarks. GRPO-Zero indicates conducting GRPO without SFT cold start like DeepSeek-R1-Zero (Guo et al., 2025). All results are obtained using 256×28×28 input resolutions and 32 frames. Implementation details are in Appendix G.1.

| Models | VKBench | MVBench | Video-MME | MMVU | VSI-Bench | Avg. |
|---|---|---|---|---|---|---|
| *Open-Source Models* | | | | | | |
| VideoLLaMA2 (Li et al., 2024a) | 55.7 | 54.6 | 47.9 | 44.8 | - | - |
| mPLUG-Owl3-8B (Wang et al., 2025b) | 64.2 | 54.5 | 53.5 | - | - | - |
| LLaVA-OneVision-7B (Li et al., 2024a) | 64.9 | 56.7 | 58.2 | 49.2 | 34.1 | 52.62 |
| *Recent R1-based Video Methods* | | | | | | |
| Video-R1-7B (Feng et al., 2025) | 65.3 | 63.9 | 59.3 | 63.8 | 30.8 | 56.62 |
| VideoRFT-7B (Wang et al., 2025b) | 64.6 | 62.1 | **59.8** | **68.5** | 35.5 | 58.10 |
| *Basemodel: Qwen2.5-VL-7B-Instruct* | | | | | | |
| Zero-Shot | 64.0 | 59.4 | 52.8 | 61.3 | 34.8 | 54.46 |
| *See-Think-Answer* SFT | 63.9 | 61.0 | 53.3 | 63.2 | 35.1 | 55.30 |
| GRPO-Zero | 64.3 | 63.8 | 58.8 | 65.6 | 23.6 | 55.22 |
| **Video-VK+** | **67.7** | **64.8** | **59.8** | 67.0 | **35.9** | **59.04** |

As shown in Table 2, our model achieves a **3.7%** improvement over the baseline on VKBench, highlighting its superior capacity to comprehend visual knowledge. Moreover, Video-VK+ demonstrates strong generalization across multiple video understanding and reasoning benchmarks, surpassing the baseline by **5.4%** on MVBench, **7.0%** on Video-MME, **5.7%** on MMVU and **1.1%** on VSI-Bench, while also outperforming previous open-source and R1-based video models. These results underscore the effectiveness of our VKQA collection and training strategy, reflecting the model's robust learning capabilities. Crucially, these findings indicate that strengthening an MLLM's grasp of visual knowledge as a bridge can consistently boost performance across both perceptual grounding and higher-level cognitive reasoning tasks.

### 5.2 ABLATION STUDY

**Contribution of Training Strategy.** As shown in Table 3, using *See–Think–Answer* SFT alone results in a notable drop in the MS and SI dimensions, likely because merely memorizing visual knowledge does not generalize to diverse and complex social scenarios. Experiments also show that both vanilla GRPO and GRPO with $r_v$ tend to repetitively describe scene phenomena in the SA

Table 3: Performance of different training strategies on VKBench. Bold indicates the best result.

| SFT | GRPO | $r_v$ | Overall | IP | OA | OM | SA | EA | MS | SR | SI |
|---|---|---|---|---|---|---|---|---|---|---|---|
| | | | 63.99 | 48.00 | 53.00 | 82.63 | 35.23 | 79.47 | 75.13 | 77.50 | 65.64 |
| ✓ | | | 63.87 | 57.00 | 54.50 | 80.00 | **43.52** | 78.95 | 69.54 | **81.67** | 58.72 |
| | ✓ | | 64.29 | 48.50 | 57.50 | 82.11 | 31.61 | 77.89 | 78.17 | 83.33 | 63.85 |
| | ✓ | ✓ | 63.63 | 53.50 | 56.50 | 82.11 | 22.28 | **81.58** | 77.16 | 80.83 | 63.08 |
| ✓ | ✓ | | 65.71 | 52.50 | 59.50 | 83.16 | 42.49 | 78.42 | 78.68 | 79.17 | 61.79 |
| ✓ | ✓ | ✓ | **67.74** | **58.00** | **60.50** | **83.68** | 38.34 | 81.05 | **79.19** | 80.83 | **66.92** |

dimension, struggling with structured output. Only the two-stage combination of SFT and GRPO achieves consistent gains across all dimensions, with $r_v$ providing an additional 2% improvement. SFT establishes a foundation for generating structured outputs, while RL enables the model to generalize across diverse visual knowledge scenarios (Chu et al., 2025). However, Video-VK+ still underperforms other baselines in SA. This is likely because it involves long-term visual memory, and MLLM augmented by $r_v$ tends to capture excessive visual knowledge without focusing on the most relevant ones, highlighting the need for better spatial visual comprehension in current MLLMs.

Table 4: Performance of different $\lambda$.

| $\lambda$ | 0.0 | 0.1 | 0.3 | 0.5 | 0.7 | 1.0 |
|---|---|---|---|---|---|---|
| VKBench | 65.96 | 66.79 | 65.71 | 65.83 | 65.77 | 64.64 |

Table 5: Performance of different verifier model.

| Verifier Model | Zero-Shot | Qwen2.5-7B | Qwen2.5-VL-7B |
|---|---|---|---|
| VKBench | 63.99 | 66.49 | 67.74 |

**Choice of Visual Knowledge Reward Ratio $\lambda$.** We study the impact of varying $\lambda$, which controls the weight of the visual knowledge reward $r_v$, by training 1K RL steps for rapid exploration. As shown in Table 4, Video-VK+'s performance on VKBench is sensitive to this parameter. The best performance **66.79** is achieved at $\lambda = 0.1$, both omitting the reward and overemphasizing it lead to noticeable drops, indicating that either reliance or disregard of visual knowledge is suboptimal. A moderate incorporation of $r_v$ is crucial for maximizing Video-VK+'s performance.

**Choice of Verifier Model.** We examined the impact of the verifier model on VKBench performance by using both Qwen2.5-VL-7B and its LLM-only counterpart, Qwen2.5-7B (Team, 2024) to compute the visual knowledge reward. As shown in Table 5, using MLLM to verify performs slightly better. Using the same verifier as the basemodel better simulates how MLLM leverages the internal visual knowledge acquired by itself. While a stronger LLM verifier model might yield higher gains, this would diverge from our goal of encouraging reliance more on the models' own visual knowledge rather than on language reasoning alone.

## 6 CONCLUSION

In this paper, we highlight the critical role of visual knowledge in the development of MLLMs, which encompasses concepts rooted in human cognitive psychology and serves to bridge perception and reasoning. To measure this gap quantitatively, we introduce **VKBench**, a multimodal benchmark designed to evaluate MLLMs' understanding of visual knowledge across both world-centric and human-centric scenarios. Furthermore, we propose **Video-VK+**, an initial attempt to explicitly integrate visual knowledge into MLLMs using reinforcement learning with **VKQA**, achieving notable performance on VKBench as well as other video benchmarks. We hope that our work could provide some insight for the development of vision-oriented MLLMs.

**Limitation.** The current VKBenchis limited to multichoice QA form. Although effective for standardized assessment, this format does not capture the full complexity of open-ended reasoning and may not fully expose the nuances of model failures. Future work could extend the benchmark to include more challenging formats, such as free-form generation and interactive dialogues. Besides, Video-VK+ builds on existing models using established techniques such as SFT and GRPO. While this demonstrates that visual knowledge can be learned even with these methods, designing new models and training objectives specifically optimized for acquiring and reasoning with visual knowledge remains a promising direction for future research.

ETHICS STATEMENT

All authors have carefully reviewed and agree to comply with the ICLR Code of Ethics throughout the research process and paper submission. The datasets employed in this study are either openly accessible benchmark resources or synthetically created exclusively for research objectives. The methods and results presented are intended to contribute to a deeper scientific understanding of machine learning models and do not entail direct risks of misuse or potential harm. We uphold the principles of fairness, transparency and accountability in our work and advocate for responsible application of the datasets and techniques introduced in this study.

REPRODUCIBILITY STATEMENT

We have taken comprehensive instructions to ensure the reproducibility of our benchmark and experimental results. Section 3.3 and Appendix E provide detailed descriptions of the data collection and filtering procedures for VKBench. Section 3.4 and Appendix G.2 outline the evaluation protocols and metrics, including the prompts and hyperparameters used to conduct evaluations on VKBench. Section 4 and Appendix F describe the training strategy for Video-VK+ and the complete VKQA generation pipeline. Section 5 and Appendix G.1 detail the experimental setup, including all hyperparameter choices and implementation details necessary to replicate our results. To further facilitate reproducibility, we will release all code, model weights, and datasets under appropriate open-source licenses in the future.

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

## A    THE USAGE OF LARGE LANGUAGE MODELS (LLMs)

In our work, LLMs were employed to assist in the preliminary screening of the data, which was subsequently reviewed and finalized by human researchers. In addition, LLMs were used to support proofreading and linguistic refinement of the manuscript. All content presented here has been rigorously verified to ensure a faithful representation of the original intent of the authors and to eliminate any factual inaccuracies or hallucinations potentially introduced by the models.

## B    DETAILED DEFINITION OF EIGHT TASKS OF VISUAL KNOWLEDGE

**Intuitive Physics.**    Intuitive Physics refers to the cognitive ability to judge the physical plausibility of dynamic events by applying a common-sense understanding of physical principles such as object permanence, solidity, and continuity (Spelke et al., 1992; Baillargeon, 1986). This goes beyond simply recognizing objects to predicting how they will behave under physical laws. For instance, if a ball rolls behind a screen, we intuitively expect it to continue its trajectory and emerge from the other side, rather than vanishing or passing through the solid screen. This core knowledge, which develops earlier than the motor skills needed to search for hidden objects, was famously demonstrated in violation-of-expectation experiments where infants looked longer at physically impossible events (Baillargeon, 1986; Piaget, 1954).

**Object Affordance.**    Object Affordance is the cognitive capacity to determine an object's potential functional uses based on its perceptual and structural properties relative to an agent's own capabilities (Gibson, 1979). It is the direct perception of action possibilities. For example, a horizontal, flat, knee-high surface affords sitting, a small, graspable object affords lifting, and a sharp edge affords cutting. This is distinct from object identification; one perceives a chair's "sittability" directly, rather than first identifying it as a chair and then deducing its function. The design of everyday objects often relies on making these affordances clear and perceivable to the user (Norman, 1988).

**Object Material.**    Object Material involves identifying the material composition of objects by leveraging visual cues such as texture, gloss, and transparency (Fleming, 2017; Adelson, 2001). This ability is crucial for predicting an object's physical properties (e.g., weight, fragility, texture) without direct tactile interaction. For example, by looking at a drinking glass, one can infer it is made of glass and is therefore rigid, fragile, and smooth, whereas a paper cup is understood to be flexible, light, and opaque. Human perception of materials is organized along both perceptual dimensions (e.g., gloss, grainy) and conceptual ones (e.g., mineral, viscous), indicating a blend of low-level visual analysis and high-level knowledge (Schmidt et al., 2025).

**Spatial Awareness.**    Spatial awareness is the cognitive understanding of objects' relative positions, orientations, and relationships within a given environment O'keefe & Nadel (1978). It involves creating a mental representation, or "cognitive map" of a scene that allows for navigation and localization. This is distinct from simply identifying objects, it is about knowing where they are in relation to one another. For example, in an image, a model with spatial awareness understands that a cat is sitting on a mat, not under it, and that a lamp is to the left of the sofa.

**Event Anticipation.**    Event Anticipation is the cognitive faculty for inferring the most probable subsequent events based on existing video clues and learned social norms or scripts (Schank & Abelson, 1977). This ability allows for proactive, rather than reactive, behavior. For example, upon seeing a person in a restaurant reading a menu, one anticipates that the next likely event is the person ordering food from a waiter, not standing up to leave. This prediction relies on a learned "restaurant script" that outlines the typical sequence of actions in that context. At a more fundamental level, the brain is viewed as a prediction machine that constantly generates and updates hypotheses to minimize prediction errors about future sensory input (Friston, 2010; Clark, 2013).

**Mental State.**    Mental State inference, often termed *Theory of Mind*, is the ability to infer the internal psychological states of others, including their beliefs, desires, intentions, and emotions, and to understand that these can differ from one's own (Premack & Woodruff, 1978; Baron-Cohen et al., 1985). A classic test involves understanding false beliefs: if Sally puts her marble in a basket and

leaves, and Anne then moves the marble to a box, a person with Theory of Mind understands that Sally will still look for the marble in the basket because she holds a false belief (Wimmer & Perner, 1983). This is distinct from simply recognizing an emotion; for example, recognizing a smile as happiness is a component, but inferring that someone is happy, because they believe they won is a Theory of Mind judgment (Ekman, 1992).

**Social Relation.** Social Relation inference is the capacity to discern interpersonal relationships and social dynamics (e.g., intimacy, dominance) from non-verbal cues. These cues include physical proximity (proxemics), gaze patterns, and body posture (Hall, 1966; Argyle & Dean, 1965). For example, observing two individuals standing very close together, maintaining frequent eye contact, and having open body postures suggests a familiar and positive relationship, such as that between close friends. In contrast, two individuals maintaining a large distance, avoiding eye contact, and using closed postures (e.g., crossed arms) would suggest a more formal or distant relationship, like that between strangers.

**Subjective Intention.** Subjective Intention involves reconstructing the underlying goals or motivations behind an observed action, moving beyond what a person is doing to infer why they are doing it (Dennett, 1987). This requires treating others as rational agents whose actions are driven by their beliefs and desires. For instance, if you see someone repeatedly failing to place a book on a high shelf, you infer their intention is to shelve the book, even though the action itself is unsuccessful. This is different from moral evaluation, which judges the rightness or wrongness of an intention; here, the task is simply to identify the goal itself (Piaget, 1965; Brentano, 1995).

## C  THEORETICAL ANALYSIS OF VIDEO-VK+

The reasoning process of MLLMs can be characterized as a conditional probability optimization grounded in the visual evidences. Given a visual input $\mathbf{V}$ (e.g., images or videos), a user query $\mathbf{T}_q$, and corresponding ground-truth anwser $\mathbf{T}_a$, we formulate the MLLMs' reasoning over these multimodal inputs as:

$$\underbrace{P(\mathbf{T}_a|\mathbf{V}, \mathbf{T}_q)}_{\text{VQA Accuracy}} \propto \underbrace{P(\mathbf{V}|\mathbf{T}_a, \mathbf{T}_q)}_{\text{Visual Evidences}} \cdot \underbrace{P(\mathbf{T}_a|\mathbf{T}_q)}_{\text{Language Priors}}, \tag{2}$$

where $P(\mathbf{V}|\mathbf{T}_a, \mathbf{T}_q)$ measures the consistency between the extracted visual representation from $\mathbf{V}$ and $\mathbf{T}_a$. We can expect a higher $P(\mathbf{V}|\mathbf{T}_a, \mathbf{T}_q)$ when those visual features embed rich and precise descriptions related to $\mathbf{T}_a$, indicating visual evidences support the answer more confidently. In contrast, $P(\mathbf{T}_a|\mathbf{T}_q)$ reflects the language prior, the model's pre-existing world knowledge encoded during LLM pretraining. When visual evidence is ambiguous or insufficient, MLLMs tend to rely more heavily on this prior $P(\mathbf{T}_a|\mathbf{T}_q)$, which may lead to plausible but incorrect, or even hallucinated responses. Accordingly, the final output of MLLMs aims to maximize the following:

$$\hat{\mathbf{T}}_a = \arg\max_{\mathbf{T}_a} P(\mathbf{V}|\mathbf{T}_a, \mathbf{T}_q) \cdot P(\mathbf{T}_a|\mathbf{T}_q). \tag{3}$$

Note it is sound to regularize how MLLMs reason in a postive manner by explicitly improving the visual evidences $P(\mathbf{V}|\mathbf{T}_a, \mathbf{T}_q)$. Strengthening this component encourages the model to ground its predictions more firmly in visual knowledge, thereby reducing over-reliance on language priors and mitigating hallucination risks.

## D  PRINCIPLE OF GROUP RELATIVE POLICY OPTIMIZATION

Group Relative Policy Optimization (GRPO) (Shao et al., 2024) has demonstrated notable effectiveness across both textual and visual tasks (Huang et al., 2025; Feng et al., 2025). GRPO estimating the baseline from group scores, significantly reducing training computation. Specifically, it samples $G$ outputs $\{o_1, ..., o_G\}$ for each query $q$ from old policy $\pi_{\theta_{old}}$, these generations are then evaluated with predefined rule-based rewards, such as format reward and accuracy reward, the final advantage value of each group $A_i$ is calculated as:

$$A_i = \frac{R_i - \text{mean}(\{R_j\})}{\text{std}(\{R_j\})} \tag{4}$$

Finally, the policy model $\pi_\theta$ will be optimized by maximizing the following objective, where $\epsilon$ and $\beta$ are hyper-parameters, and $\pi_{ref}$ is the reference policy:

$$
\mathcal{J}_{\text{GRPO}}(\theta) = \mathbb{E}_{[q,\{o_i\}]} \left[ \frac{1}{G} \sum_{i=1}^{G} \left[ \min\left( \frac{\pi_\theta}{\pi_{\theta_{\text{old}}}} A_i, \text{clip}\left( \frac{\pi_\theta}{\pi_{\theta_{\text{old}}}}, 1-\epsilon, 1+\epsilon \right) A_i \right) - \beta \mathbb{D}_{\text{KL}}(\pi_\theta || \pi_{\text{ref}}) \right] \right]
$$
(5)

## E  DETAILS OF VKBENCH

### E.1  DATA SOURCE

We summarize the annotation sources and video sources for each task in VKBench. The collection spans diverse scenarios and provides reasonably accurate initial annotations, which serve as the foundation for constructing VKBench.

Table 6: Data source of VKBench.

| Task | Annotation Source | Video Source |
|---|---|---|
| Intuitive Physics | IntPhys2 (Bordes et al., 2025) | Unreal Engine (Epic Games, 2019) Simulated |
| Object Affordance | PACS (Yu et al., 2022) | YouTube |
| Object Material | PACS (Yu et al., 2022) | YouTube |
| Spatial Awareness | VSI-Bench (Yang et al., 2025b) | ARKitScenes (Baruch et al., 2021) |
| Event Anticipation | VLEP (Lei et al., 2020) | YouTube Lifestyle Vlog clip |
| Mental State | Social-IQ-2.0 (Wilf et al., 2023) | YouTube |
| Social Relation | Social-IQ-2.0 (Wilf et al., 2023) | YouTube |
| Subjective Intention | RexTime (Chen et al., 2024) | QVHighlights (Lei et al., 2021), ActivityNet (Caba Heilbron et al., 2015) |

### E.2  STATISTICAL CHARACTERISTICS

The statistical characteristics of our collected dataset are illustrated in Figure 6. Compared with other tasks, videos in *Spatial Awareness* and *Subjective Intention* tend to have longer durations, providing richer visual knowledge for MLLMs to capture and thereby increasing the level of difficulty.

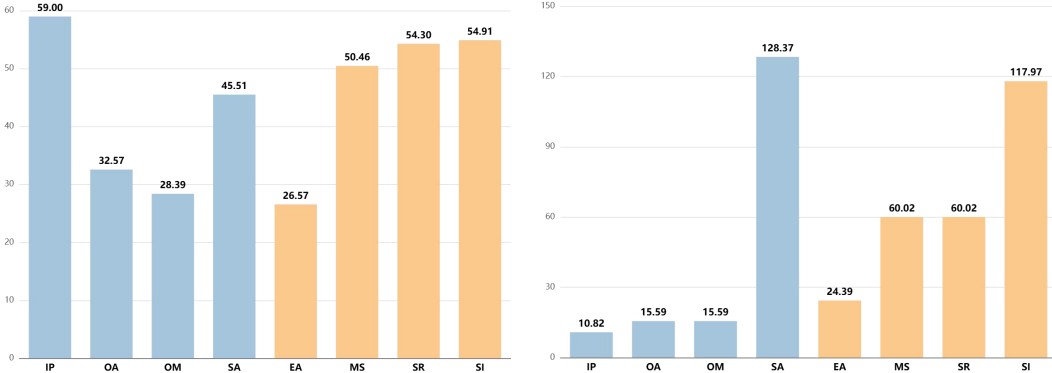

Figure 6: Statistical characteristics of per task in VKBench. **Left**. Average lengths of question and options (in words). **Right.** Average video duration (in seconds).

### E.3 QA FILTRATION

The text-only LLM accuracy on VKBench QA pool is shown in Figure 7. Through progressive filtering (Section 3.3) with text-only LLMs, combined with DeepSeek-R1 rewriting of unsuitable options, the accuracy of text-only models under the blind QA setting steadily decreases. For instance, Qwen3-32B-Thinking drops from **55.1%** to **34.7%**, approaching the random-choice level. This trend demonstrates the effectiveness of our QA filtration process in substantially mitigating language bias, thereby preventing MLLMs from exploiting linguistic shortcuts and ensuring that they must genuinely rely on visual knowledge to answer questions in our VKBench.

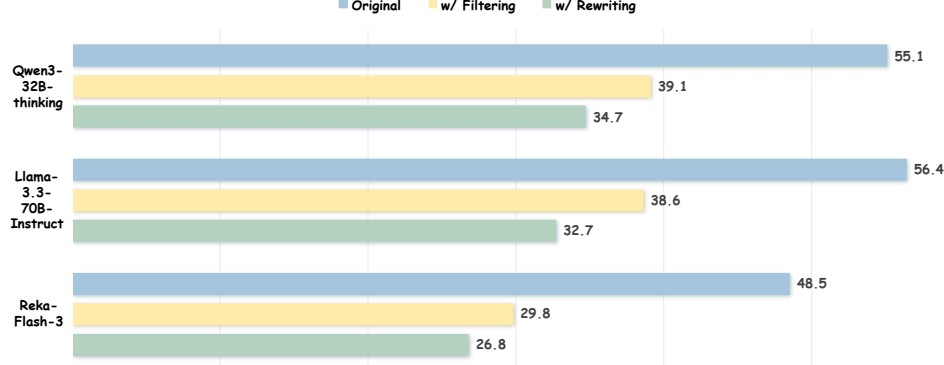

Figure 7: Text-only LLM accuracy on VKBench QA pool.

The prompt provided to DeepSeek-R1 to enhance options is illustrated in Figure 8.

---

**Rewriting Options Prompt**

You are an expert MCQ (Multiple Choice Question) evaluator and editor tasked with assessing and improving question options.
Question:{question}
Current Options: A) ... B) ... C) ... D) ...
Correct Answer: {answer}

**Task**:
1. Rate the options' quality (1 = good, 0 = needs improvement) based on:
   - Distinctiveness of each option
   - Plausibility of distractors
   - Relevance to question
   - Avoidance of redundancy/nonsense
   - Appropriate difficulty level
2. If rating is 0, provide improved options following above guidelines and take into account the following:
   - Preserve the correct answer's validity
   - Ensure all distractors are plausible but incorrect

**Output Requirements:**
You MUST provide your evaluation in the following strict JSON format:
{{"rating": 0,"improved_options_1": "Revised option 1",
"improved_options_2": "Revised option 2", "improved_options_3": "Revised option 3","improved_options_4": "Revised option 4"}}
or {{"rating": 1}}

Important Notes:
1. Output must be valid JSON (check for proper quotes, commas, etc.)
2. Do not output any other explanation or comment.

---

Figure 8: Prompt for DeepSeek-R1 (Guo et al., 2025) to enhance options.

## F    DETAILS OF VKQA

### F.1    VKQA-30K DATA SOURCES

Table 7: Data source of VKQA-30K.

| Category | Source | Size (%) |
|---|---|---|
| General | LLaVA-Video-178K (Zhang et al., 2024c), Video-R1-260K (Feng et al., 2025) NextQA (Xiao et al., 2021) | 12K (40.0%) |
| World-Centric | CLEVRER (Yi et al., 2019), VSI-100K (Liao et al., 2025), Intphys (Riochet et al., 2018), STAR (Wu et al., 2024) | 9K (30.0%) |
| Human-Centric | EMER (Lian et al., 2023), MAFW (Liu et al., 2022), Social-IQ (Zadeh et al., 2019), CausalVidQA (Li et al., 2022) | 9K (30.0%) |

### F.2    STATISTICAL CHARACTERISTICS

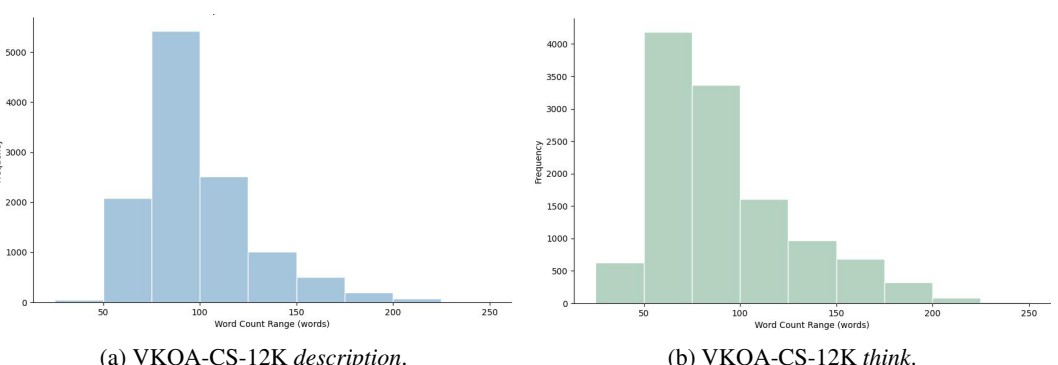

(a) VKQA-CS-12K *description*.          (b) VKQA-CS-12K *think*.

Figure 9: Word length distribution visualizations of VKQA.

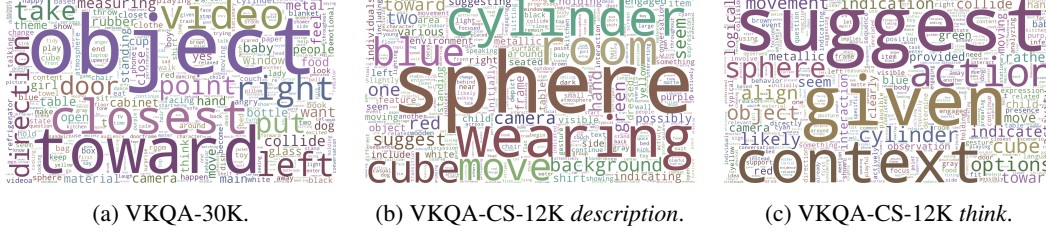

(a) VKQA-30K.          (b) VKQA-CS-12K *description*.          (c) VKQA-CS-12K *think*.

Figure 10: Word cloud visualizations of VKQA.

### F.3    VKQA-CS-12K GENERATION PIPELINE

Starting from the original VKQA-30K collection, we first prompted an MLLM (Qwen2.5-VL-7B-Instruct) to produce *See-Think-Answer* style responses, as outlined in Figure 12. Only those question–answer pairs where the model's output was both accurate and matched the required format were kept. In the next stage, we applied an additional filtering step using the prompt shown in Figure 13. Here, we preserved only the instances that the model could correctly solve when provided with textual descriptions of their visual content alone generated before, thereby ensuring that the reasoning relied strictly on explicit visual cues rather than implicit priors. After these two refinement steps, we obtained a curated subset of about 12,000 high-quality QA pairs, which we denote as VKQA-CS-12K.

# G    IMPLEMENTATION DETAILS

## G.1    TRAINING

**Baselines.**    Video-R1 (Feng et al., 2025) and VideoRFT (Wang et al., 2025b) are employed for comparisons. The former explores the R1 paradigm for video reasoning with MLLMs, with a SFT cold start and following T-GRPO manner, while the latter follows the similar training style, incorporating higher quality COT data and semantic-consistency reward to guide the reasoning trajectories.

**Prompts.**    We use the prompt shown in Figure 12 to guide the model in generating *See-Think-Answer* responses during both SFT and GRPO. Prompt shown in Figure 13 is used to calculate the binary visual knowledge reward with the help of regular expression matching.

**Hyper-parameters.**    Our training implementation is based on TRL framework (von Werra et al., 2020). All hyper-parameters we use for the main experiments are reported in Table 8. To balance training efficiency and computational resource constraints, we adopt the strategy from Huang et al. (2025); Wang et al. (2025b) by limiting the maximum number of video frames at 16 during training. Each frame is processed at a resolution of up to $128 \times 28 \times 28$ pixels. All experiments are reproducible using 8 NVIDIA A800 GPUs (80GB).

Table 8: Training hyperparameters.

| Parameter | SFT | GRPO |
|---|---|---|
| train type | full | full |
| use vllm | false | true |
| vllm gpu memory utilization | - | 0.7 |
| attn impl | flash_attn2 | flash_attn2 |
| deepseed config | zero2 | zero3 |
| torch dtype | bfloat16 | bfloat16 |
| num train epochs | 1 | 1 |
| per device train batch size | 1 | 1 |
| gradient accumulation steps | 2 | 1 |
| num generations | - | 8 |
| KL coefficient $\beta$ | - | 0.04 |
| visual knowledge reward ratio $\lambda$ | - | 0.1 |
| learning rate | 1e-6 | 1e-6 |
| max prompt length | 16384 | 16384 |
| max completion length | 1024 | 1024 |
| max grad norm | 5.0 | 5.0 |
| min frames | 4 | 4 |
| max frames | 16 | 16 |
| video pixels | $128 \times 28 \times 28$ | $128 \times 28 \times 28$ |

## G.2    EVALUATION

The models evaluated in our study vary significantly in both architecture and scale. All experiments are conducted on NVIDIA A800 GPUs with 80 GB of memory. To ensure the fidelity and reproducibility of our results, we strictly follow the official implementations and configurations released by the model developers. Detailed specifications of the evaluated models are summarized in Table 9.

**Prompts.**    For models that do not require thinking, we use the prompt in Figure 11 to guide their concise responses. For Video-VK+, we use the same prompt in Figure 12 as in the training stage. For other thinking models, we follow the official prompts specified by their developers.

**Hyper-parameters.**    For QwenVL-based models, in particular QwenVL-2.5, MiMo-VL, and our Video-VK+, we perform evaluation with the number of video frames ranges from 4 to 32, while the video resolution is fixed at $256 \times 28 \times 28$ pixels. For other open-source and proprietary models, we fix the maximum number of input video frames to 32 to ensure comparability. Except for proprietary models, which are evaluated with a temperature of 1.0, all open-source models are evaluated with a temperature of 0.1 and a top_p value of 0.001.

Table 9: Details of evaluated multimodal large language models used in VKBench.

| Organization | Model | Release | Version |
|---|---|---|---|
| ***Proprietary MLLMs*** | | | |
| OpenAI | GPT-4o | 2024-11 | `gpt-4o-2024-11-20` |
| Google | Gemini-2.5-Flash | 2025-5 | `Gemini-2.5-Flash` |
| | Gemini-2.5-Pro | 2025-3 | `Gemini-2.5-Pro` |
| ***Open-source MLLMs*** | | | |
| DAMO-NLP | VideoLLaMA2-7B | 2024-6 | `VideoLLaMA2-7B` |
| LMMs-Lab | LLaVA-OV-7B | 2024-8 | `LLaVA-OV-7B` |
| | LLaVA-OV-72B | 2024-8 | `LLaVA-OV-72B` |
| | LLaVA-Video-7B | 2024-10 | `llava-onevision-qwen2-7b-ov` |
| | LLaVA-Video-72B | 2024-10 | `llava-onevision-qwen2-72b-ov-chat` |
| mPLUG | mPLUG-Owl3-7B | 2024-11 | `mPLUG-Owl3-7B-241101` |
| OpenBMB | MiniCPM-V-2.6 | 2024-8 | `MiniCPM-V-2_6` |
| | MiniCPM-V-4.5 | 2025-9 | `MiniCPM-V-4_5` |
| Alibaba | Qwen2.5-VL-3B | 2025-1 | `Qwen2.5-VL-3B-Instruct` |
| | Qwen2.5-VL-7B | 2025-1 | `Qwen2.5-VL-7B-Instruct` |
| | Qwen2.5-VL-32B | 2025-1 | `Qwen2.5-VL-32B-Instruct` |
| | Qwen2.5-VL-72B | 2025-1 | `Qwen2.5-VL-72B-Instruct` |
| ZhipuAI | GLM-4.1V-9B-Thinking | 2025-7 | `GLM-4.1V-9B-Thinking` |
| Xiaomi | MiMo-VL-7B-RL | 2025-8 | `MiMo-VL-7B-RL-2508` |
| OpenGVLab | InternVL3.5-8B | 2025-9 | `InternVL3.5-8B` |
| | InternVL3.5-30B-A3B | 2025-9 | `InternVL3.5-30B-A3B-Instrcut` |
| | InternVL3.5-38B | 2025-9 | `InternVL3.5-38B` |
| | InternVL3.5-241B-A28B | 2025-9 | `InternVL3.5-241B-A28B-Instruct` |

---

## Vanilla Evaluation Prompt

Please provide only the single option letter (e.g., A, B, C, D, E,etc.) within the <answer> </answer> tags.

Figure 11: Prompt for VKBench evaluation for vanilla models.

---

## *See-Think-Answer* Prompt

You are tasked with analyzing an video to generate a detailed description to help you answer the question.

First analyze the video and produce a self-contained description— detailed enough that can lead to the correct answer. Wrap the entire description between <description> </description> tags.

Next, engage in an internal dialogue and include self-reflection or verification in your reasoning process. Provide your detailed, step-by-step reasoning based on the video description information and video, and enclose this part between <think> </think> tags.

Finally, provide only the single option letter (e.g., A, B, C, D, E, etc.) between the <answer> </answer> tags.

The output format should be: <description> video description here </description><think> reasoning process here </think><answer> answer here </answer>.

Figure 12: Prompt for MLLMs to generate to *See-Think-Answer* output format.

> **Visual Knowledge Reward Prompt**
>
> You are provided a text description of a problem and a question.
> Determine the answer to the question based on the text description.
> Provide only the single option letter (e.g., A, B, C, D, E, etc.)
> between the <answer> </answer> tags.
> The output format should be: <answer> answer here </answer>.

Figure 13: Prompt for verifier model to cauculate visual knowledge reward.

## H MORE EVALUATION ANALYSIS ON VKBENCH

### H.1 FULL PERFORMANCE OF ABLATION STUDY ON THE VERIFIER MODEL

Table 10: Full performance of different verifier models on VKBench. Bold indicates the best result.

| Verifier Model | Overall | IP | OA | OM | SA | EA | MS | SR | SI |
|---|---|---|---|---|---|---|---|---|---|
| Qwen2.5-7B | 66.49 | 51.00 | 59.50 | 83.16 | **39.38** | 79.47 | 78.68 | 80.00 | 66.67 |
| Qwen2.5-VL-7B | **67.74** | **58.00** | **60.50** | **83.68** | 38.34 | **81.05** | **79.18** | **80.83** | **66.92** |

As shown in Table 10, integrating the MLLM verifier leads to consistent performance improvements across most tasks. Although a slight drop is observed in the *Spatial Awareness*, the multimodal verifier still exhibits more balanced and robust performance overall, particularly in *Intuitive Physics* and *Event Anticipation*. These results highlight that leveraging visual signals and knowledge during verification can significantly enhance model reliability on VKBench.

### H.2 EFFECT OF FRAMES

We examine the sensitivity of model performance to the number of input frames on VKBench. As shown in Figure 14, performance varies across tasks with changing frames. Tasks such as SR and SI benefit from more frames, indicating richer visual knowledge extraction. However, some tasks (e.g., SA) show non-monotonic trends, suggesting more frames does not always yield better results, while redundant frames may introduce noise and interfere with visual knowledge. These findings highlight that making the selection of informative and critical frames, similar to how humans focus on key visual knowledge, a promising direction for future improvements.

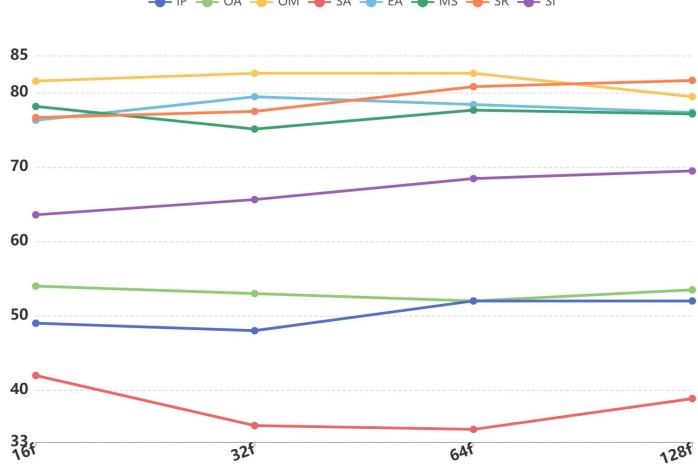

Figure 14: Performance across varying numbers of frames for each task of VKBench (Base model: Qwen2.5-VL-7B-Instruct; Input video frame pixels: $256 \times 28 \times 28$).

