# OpenReview forum: "Benchmarking Visual Knowledge in Multimodal Large Language Models"
_ICLR.cc/2026/Conference — ICLR 2026 Conference Withdrawn Submission_

### Official Review · Reviewer_em2a · 2025-10-23

**Soundness:** 3
**Presentation:** 2
**Contribution:** 2
**Rating:** 4
**Confidence:** 3

**Summary:**

The paper introduces VKBench, a benchmark comprising 1,680 questions across 1,249 videos, designed to evaluate eight core dimensions of visual knowledge spanning both world-centric and human-centric domains. It further presents VKQA, a new dataset, and Video-VK+, a baseline model that explicitly incorporates visual knowledge into multimodal large language models, which follows a structured See–Think–Answer paradigm and employs reinforcement learning with visual-knowledge rewards. Experimental results demonstrate that this approach improves performance on VKBench by 3.7% and surpasses existing models on multiple video understanding benchmarks.

**Strengths:**

1. The authors present not only VKBench, a comprehensive benchmark, but also Video-VK+, a novel method designed to evaluate and enhance visual knowledge in multimodal large language models. Their study provides systematic insights into how models apply world-centric and human-centric visual knowledge.

2. By explicitly incorporating visual knowledge through a structured See–Think–Answer framework and reinforcement learning, Video-VK+ achieves substantial performance gains on VKBench and other video understanding tasks.

**Weaknesses:**

1. Some of the presented examples in VKBench are ambiguous or confusing. See "Questions" part.

2. Human-centric tasks in VKBench have already been well studied in prior benchmarks such as Visual Commonsense Reasoning (VCR), which somewhat diminishes the novelty of this contribution.

3. The definition of visual knowledge remains unclear. According to Section 3.1, it encompasses both visual commonsense and expert-level domain knowledge, while the paper mainly focuses on the former. It may therefore be more appropriate to reframe the work around visual commonsense rather than the broader concept of visual knowledge.

**Questions:**

1. Questions about cases in Figure 2:

-  For the Object Affordance and Object Material tasks, the cases appear solvable using only a pair of images rather than a full video. The necessity of providing a video input for these tasks should therefore be further justified.

-  In the Spatial Awareness task, it is unclear how one can infer the spatial relationship between the TV and the sofa solely from three separate images of the stove, TV, and sofa, together with the description “I am standing by the stove and facing the sofa.” The rationale for how these inputs support spatial reasoning should be clarified.

2. Why is it necessary to use video as input for certain tasks? Given that many examples appear solvable with one or few static images, whether single or multi-image inputs (like VCR) would be sufficient to capture the required visual information and reasoning? What additional advantages can temporal continuity in videos provide?

---

> ### Author Response · Authors · 2025-11-14
>
> We sincerely thank the reviewer for the careful reading. Below, we address concerns in detail.
>
> **Question 1:** We apologize for the lack of clarity in Figure 2. The figure was intended as a static illustration of representative question types. Due to space limits and the difficulty of visually conveying dynamic video scenes in a static figure, we simplified the examples, which may have caused misunderstanding. For the Object Affordance and Object Material tasks, the video format is essential because visual knowledge in these domains often arises from object–object or human–object interactions. for example, how an object deforms when grasped, reflects light when rotated, or responds to applied force. Such cues cannot be inferred from isolated static frames. For the Spatial Awareness case, the underlying video captures a continuous camera sweep that reveals the stove, sofa, and TV within a closed indoor scene, allowing models to infer their relative spatial layout. ​We agree that the current static presentation does not convey this context well, and we will clarify it in the final version. We also plan to release the dataset with full video examples to better illustrate these interactions.
>
> **Question 2:** As we clarified above,  we adopt video as the input modality because much of evidences in these tasks inherently rely on temporal continuity. Static images miss these dynamic signals and often lead to appearance-based shortcuts, whereas videos preserve the rich and essential visual knowledge needed for faithful evaluation.
>
>
> **Weakness 2&3:**
> Our work is indeed inspired by VCR[1], but it is substantially different in both scope and design.
>
> 1. VCR focuses on static-image, human-centric commonsense reasoning, whereas VKBench expands to eight dimensions of visual knowledge, covering human-centric and world-centric. By leveraging videos rather than images, VKBench evaluates dynamic physical and causal understanding that VCR cannot assess.
>
> 2. Our multi-stage filtering pipeline rigorously removes audio cues and language shortcuts, making VKBench far more robust against textual or linguistic biases.
>
> 3. VKBench is explicitly designed for modern MLLMs, and it reveals fundamental weaknesses that VCR cannot expose.
>
> Importantly, our goal is not to replicate prior human-centric benchmarks like VCR. Instead, we aim to establish a unified framework for visual knowledge that jointly evaluates world-centric and human-centric understanding.
>
> [1] Zellers, Rowan, et al. "From recognition to cognition: Visual commonsense reasoning." Proceedings of the IEEE/CVF conference on computer vision and pattern recognition. 2019.

---

### Official Review · Reviewer_gT1D · 2025-10-31

**Soundness:** 3
**Presentation:** 4
**Contribution:** 2
**Rating:** 4
**Confidence:** 3

**Summary:**

This paper proposes the concept of "visual knowledge", referring to the intuitive understanding of the physical world and social knowledge that MLLMs lack. To evaluate this ability systematically, the author constructed a VKBench video benchmark, covering 8 tasks such as intuitive physics and subjective intentions. The evaluation found that the leading model lags behind humans by 15% overall, especially in physical reasoning where the gap is significant. To address this issue, the paper introduces the VKQA dataset and Video VK+ baseline model. The model adopts an "see think answer" reasoning format and incorporates visual knowledge rewards through reinforcement learning, achieving a 3.7% improvement on VKBench and demonstrating excellent generalization ability on multiple video benchmarks.

**Strengths:**

- 1. Carefully designed benchmark construction: The construction process of VKBench is very rigorous, not simply stacking existing data. The progressive filtering pipeline adopted (Minimize Audio Reliance ->Reduce Language Bias ->Enhance Wrong Candidates ->Human Verification) effectively removes language bias and audio dependencies, ensuring that the benchmark truly evaluates the model's ability to extract knowledge from visual signals, rather than its internal language model's prior knowledge.
- 2. Large scale evaluation and analysis: The paper comprehensively evaluated 23 mainstream open-source and closed-source MLLMs, not only revealing the overall performance gap, but also conducting detailed analysis (such as differences between open-source and closed-source models, model scale effects, and the impact of reasoning version). Especially through correlation analysis, it was found that the world center and human center knowledge form two independent clusters, which verifies the rationality of their classification from the data.
- 3. The paper shifts the research focus from traditional "perception" (object recognition) and "complex reasoning" to the intermediate layer of "visual knowledge", which is a relatively unexplored but crucial field. This evaluation and training data are meaningful to the MLLM community.

**Weaknesses:**

- 1. The coverage of "visual knowledge" may still be incomplete: although the division of 8 dimensions is quite comprehensive, "visual knowledge" itself is an extremely rich concept. For example, the assessment of deeper cognitive abilities such as causality, counterfactual thinking, and visual metaphor may not have been fully covered in the proposed benchmark.
- 2. VKBench only uses multiple-choice questions. This cannot fully evaluate whether the reasoning process behind it truly conforms to logic and is based on correct visual knowledge.
- 3. During the RL training of the Video VK+model, an external frozen MLLM as the verifier is introduced, which is equivalent to handing over a key evaluation criterion to another potentially flawed model. Meanwhile, this significantly increases the complexity and computational cost of training.
-4. The optimization goal of the Video VK+model is to make the visual description contain answer information as much as possible, thereby reducing the inference burden of LLM. However, it may introduce new risks: will this suppress the model from performing necessary and complex multi-step inference? Overemphasizing the "self-contained" visual knowledge may make models "lazy" for deep reasoning.
- 5. The analysis of some experimental phenomena is relatively general, such as the performance of the InternVL3.5-38B-Think on VKBench, which is actually worse than InternVL3.5-38B due to "excessive reasoning". Video VK+performs poorly on Spatial Awareness tasks, even inferior to some baseline models. The paper simply attributes the task to requiring 'long-term visual memory'. These explanations are somewhat vague and deserve a deeper analysis.

**Questions:**

Refer to Weaknesses

---

> ### Author Response · Authors · 2025-11-14
>
> We sincerely thank the reviewer for the careful reading. Below, we address concerns in detail.
>
> **Weakness 1:**
> We fully agree that visual knowledge is an inherently broad and hierarchical concept. VKBench aims to establish a foundational layer by covering eight core dimensions that bridge perception and reasoning, such as intuitive physics, spatial awareness, object affordance, and social intention, which are fundamental prerequisites for higher-level cognition.As discussed in Section 3.2, these dimensions already capture the key mechanisms of human-like visual understanding. In future work, we will consider extending VKBench to toward deeper cognitive aspects, including causal reasoning, counterfactual prediction, and visual metaphor understanding, building on this solid foundation.
>
> **Weakness 2:**
> The multiple-choice design is deliberate, following established multimodal benchmarks such as MMMU[1], MVBench[2], Video-MME[3], MMVU[4] and so on, as it allows for controlled evaluation and high reproducibility while minimizing ambiguity from open-ended responses.
>
> **Weakness 3&4:**
> Deep reasoning is not always necessary because our VKBench involves relatively straightforward visual knowledge without expert-level knowledge. The design of Video-VK+ is to explicitly encourage the model to rely on self-contained visual knowledge rather than a language shortcut. As analyzed in Section 3.4, ''Thinking or Not'', InternVL3.5-38B-Think drops in performance due to severe repetition during reasoning.  Overly long reasoning chains introduce noise rather than useful insight. This indicates that a proper level of language reasoning is essential; too little fails to integrate visual knowledge, while too much becomes counterproductive.
>
> **Weakness 5:**
> We appreciate this insightful suggestion. To further clarify the benchmark’s diagnostic value, we will include qualitative analyses in the revised version, showcasing representative examples of model failures on VKBench to illustrate how and where current MLLMs struggle with visual knowledge understanding.
>
> [1] Yue, Xiang, et al. "Mmmu: A massive multi-discipline multimodal understanding and reasoning benchmark for expert agi." Proceedings of the IEEE/CVF Conference on Computer Vision and Pattern Recognition. 2024.
>
> [2] Li, Kunchang, et al. "Mvbench: A comprehensive multi-modal video understanding benchmark." Proceedings of the IEEE/CVF Conference on Computer Vision and Pattern Recognition. 2024.
>
> [3] Fu, Chaoyou, et al. "Video-mme: The first-ever comprehensive evaluation benchmark of multi-modal llms in video analysis." Proceedings of the Computer Vision and Pattern Recognition Conference. 2025.
>
> [4] Zhao, Yilun, et al. "Mmvu: Measuring expert-level multi-discipline video understanding." Proceedings of the Computer Vision and Pattern Recognition Conference. 2025.

---

### Official Review · Reviewer_LbT3 · 2025-11-01

**Soundness:** 3
**Presentation:** 3
**Contribution:** 3
**Rating:** 6
**Confidence:** 4

**Summary:**

Current MLLMs can see but often don’t really understand the physical/social structure of the world. So they define that missing layer as “visual knowledge”, a zone between pixels and reasoning (gravity, affordances, etc). They build VKBench: 1,249 videos, 1,680 MCQs, 8 dimensions split into world-centric (physics, affordance, material, spatial) and human-centric (event anticipation, mental state, social relation, intention). They very deliberately filter out audio and language shortcuts so models can’t just guess from text. Then they show: even strong video MLLMs are worse than humans overall, and the gap is especially bad on world-centric stuff (intuitive physics, spatial). To show this isn’t hopeless, they build Video-VK+: basically Qwen2.5-VL-7B with a See–Think–Answer format + GRPO RL + a visual-knowledge reward that checks whether your description was good enough to answer.

**Strengths:**

1. Proposes 8 tasks that map neatly to cognitive/vision literature.
2. Complete anti-shortcut pipeline.
3. The papers shows models are OK on human-centric but bad on world-centric.
4. The benchmark is balanced and well-scoped.

**Weaknesses:**

1. A lot of the difficulty comes from filtering existing datasets, not from filming new, adversarial, physics-centric videos, leading to any upstream video biases it originally uses.
2. The visual-knowledge reward uses a frozen MLLM as verifier. That’s convenient, but it bakes the verifier’s biases right back into training.
3. The paper shows correlations, but not what models actually get wrong?

**Questions:**

Some tasks (event anticipation, social relation) clearly like longer context. Why did you fix at 32 instead of reporting a “long-context” track?

---

> ### Author Response · Authors · 2025-11-14
>
> We sincerely thank the reviewer for the careful reading and the encouraging feedback. Below, we address concerns in detail.
>
> **Weakness 1:**
> We used existing datasets mainly because their annotations are of high quality and well-validated. Moreover, through our multi-stage filtering pipeline (see Section 3.3, Appendix E), we removed linguistic and statistical biases, ensuring that the remaining questions truly test visual-knowledge reasoning rather than artifacts from upstream data.
>
> **Weakness 2:**
> We note that the verifier does not judge semantic correctness, but only checks whether the See–Think part contains sufficient visual evidence to support the answer. Thus it acts as a structural validator rather than a knowledge teacher, which greatly limits the propagation of its own biases. We also deliberately use a verifier with similar capacity to the policy model to avoid bias amplification. Empirically, swapping the MLLM verifier with an LLM-only verifier changes accuracy by only ~1%, showing that the reward is largely insensitive to verifier bias. Moreover, our method encourages stronger visual grounding, which naturally reduces reliance on language priors and counteracts potential bias. A straightforward way to further mitigate verifier bias, if needed, is to use an ensemble of (M)LLMs as verifiers. However, this would substantially increase training cost and system complexity. Since our goal is primarily to evaluate visual knowledge and make an initial attempt at injecting visual knowledge into MLLMs, we leave such more complex designs to future work.
>
> **Weakness 3:**
> We appreciate this insightful suggestion. To further clarify the benchmark’s diagnostic value, we will include qualitative analyses in the revised version, showcasing representative examples of model failures on VKBench to illustrate how and where current MLLMs struggle with visual knowledge understanding.
>
> **Question 1:**
> We fix the frame number at 32 for QwenVL-based models to align with Video-R1 and Video-RFT, ensuring fair comparison with prior work. Moreover, as discussed in Appendix H.2, we conducted a detailed sensitivity analysis on the number of input frames. As shown in Figure 14, performance indeed varies by task: tasks like Social Relation and Subjective Intention benefit from longer context, while Spatial Awareness shows non-monotonic trends where excessive frames introduce noise. These findings suggest that simply increasing frame count does not guarantee improvement. Selecting informative and critical frames, akin to human visual attention, is a promising direction for future extensions of VKBench.

---

### Official Review · Reviewer_XciW · 2025-11-02

**Soundness:** 3
**Presentation:** 4
**Contribution:** 2
**Rating:** 2
**Confidence:** 4

**Summary:**

This paper identifies a core gap in MLLM's ability to do human-like visual reasoning, which they term as _visual knowledge_. This refers to intuitive principles which humans use freely to understand the world, like intuitive physics and social cues. Based on this, the paper proposes two main contributions:
1. A new benchmark, **VKBench**, which is curated from a set of existing datasets (IntPhys 2, PACS, VSI-Bench, VLEP, Social-IQ 2.0, RexTime) to benchmark MLLM's visual knowledge across world and human-centric axes. It consists of 1,680 multiple-choice questions across 1,249 videos, covering eight distinct types of visual knowledge. It was carefully constructed to avoid audio and linguistic biases.
2. A new dataset, **VKQA**, of visual knowledge video examples, as well as method, **Video-VK+**, demonstrating that visual knowledge can be taught to these models. The authors utilize a "See-Think-Answer" format with RL to enforce the model to first visually process the input before making deductions. This is done on Qwen-2.5-VL-7B Instruction backbone, and this generally improves the models by 4.58% on average on their suite of benchmarks.

The authors benchmark many state of the art models on **VKBench**, and find that models still fall behind human performance (15% at best), especially at Intuitive Physics and Spatial Awareness.

**Strengths:**

**Well-curated benchmark methodology**: VKBench was collected from existing datasets, so while there is no new contribution of base data on this front, there was consideration to how biased questions could be to audio and language, which are important problems present in recent benchmarks. There was also human validation done on the questions, in addition to shuffling the choices.

**Simultaneous proposal of problem, benchmark, and method**: The authors not only codify a problem, but they also propose a benchmark, dataset, and method to solve it. This is a fair undertaking which shows that the overarching paper was thought out well in advance.

**Nice figures and writing**: The figures are colorful and illustrative, and supplement the main text very well. Overall the writing of the paper is clear apart from minor details not described.

**Weaknesses:**

**Benchmark is not difficult nor prescriptive**: While the benchmark claims to evaluate visual knowledge, it's not clear to me what information I gain by benchmarking my model on VKBench. First, benchmarks currently introduced where SoTA models achieve 71% accuracy is not helpful, as I suspect such a gap can be closed quite quickly (especially as it seems to be implied that this is a knowledge issue from the methods section). Given that random chance is so high for many questions, I find it surprising the authors didn't add in more options or leave the questions as short answers. This to me also points to how the current division of visual knowledge axes is too coarse, as models are somewhat uniform across them, and I can't tell where a model specifically lacks when it performs poorly. I would like more feedback on where models tend to fail, or _how they do_, rather than attributing it to being a knowledge or processing issue.

Rather than looking at Pearson correlations within the benchmark itself, you should compare to other benchmarks to see how well being good at X task correlates with having a strong world simulator in your model, or social understanding (of which there exist other benchmarks already). This would provide stronger validation that this benchmark is indeed prescriptive.

**Weak method contributions beyond GRPO-Zero**: From what I can tell, it seems that the majority of the contribution of the method lies with GRPO(-Zero), which is prior work. For instance, MVBench, Video-MME, and MMVU are very similar between GRPO-Zero and Video-VK+, while I suspect VKBench may be more knowledge-based as a benchmark, which is why it improves better when incorporating extra data from VKQA. In fact, the See-Think-Answer SFT does not improve VKBench at all, while only providing modest contributions for the rest of the benchmarks.

**Questions:**

1. It's not quite clear to me from the text if the QAs are also pulled from the existing datasets, or if only the annotations are used to then synthesize new QAs. Clarification on this would be helpful.

I think the dataset needs to go under significant revisions for this to be helpful for the community. While there clearly has been a large amount of detailed effort invested, I don't think that this benchmark is of critical value given the already immense number of video benchmarks which exist. It seems to be in a niche which is already covered by other benchmarks (given that the questions are collected from other popular datasets), and even the filtering done does not have validation to show that this provides a meaningful improvement over alternatives. Therefore I recommend a reject based on my current review of the paper.

---

> ### Author Response · Authors · 2025-11-14
>
> We sincerely thank the reviewer for the careful reading. Below, we address concerns in detail.
>
> **Weakness 1:**
> 1. VKBench is not designed to be “hard” in an absolute sense, but to systematically isolate and measure “visual knowledge”, a capability bridging perception and reasoning that has been largely overlooked. We respectfully disagree that SoTA performance implies the benchmark is easy or that the gap can be quickly closed. Even the strongest model still lags human performance by over 15% overall and by 30–40% on world-centric dimensions such as Intuitive Physics and Spatial Awareness. These persistent gaps are consistent across architectures (Qwen2.5-VL, GLM-4.1V, LLaVA-72B, InternVL3.5) and training paradigms (supervised vs. RL), revealing a systematic deficiency in visual knowledge understanding, rather than an optimization artifact. Therefore, VKBench’s value lies not in its raw difficulty but in its ability to evaluate how current MLLMs fail to internalize visual knowledge, the core competence underlying world-centric and human-centric understanding, the very phenomenon it is designed to assess.
> 2. The issue of random-chance baselines should indeed be discussed by task dimension, as detailed in Section 3.3. (1) Human-centric tasks. We have already adopted a multi-stage filtering pipeline (see Appendix E.3) to remove any question–answer pairs that could be solved through textual priors or audio correlations. After this multi-step filtering, the model accuracies on human-centric tasks dropped substantially (see Figure 7), indicating that the residual items indeed require genuine visual–social understanding rather than guessable cues.(2) World-centric tasks. We intentionally preserved the binary or two-choice structure from the original high-quality source datasets (e.g., Intuitive Physics uses “possible” vs. “impossible”; Object Affordance and Object Material require selecting between two candidate objects across paired video clips). Although the number of options is smaller, these questions are unambiguous and have clearly defined ground-truth answers, making them ideal for isolating visual-knowledge reasoning rather than linguistic interpretation. Expanding the distractor set in such cases would likely introduce artificial ambiguity without improving diagnostic validity.
> 3. We appreciate this insightful suggestion. To further clarify the benchmark’s diagnostic value, we will include qualitative analyses in the revised version, showcasing representative examples of model failures on VKBench to illustrate how and where current MLLMs struggle with visual knowledge understanding.
> 4. We will also incorporate cross-benchmark correlation analysis in the revision. We agree that cross-benchmark correlation analysis would provide stronger evidence for the prescriptive validity of VKBench.
>
> **Weakness 2:**
> We respectfully disagree with the conclusion that the majority of the contribution of the method lies with GRPO(-Zero). While our method builds on the GRPO framework, its biggest gains come from explicitly embedding visual knowledge via the See–Think–Answer format and the visual-knowledge reward $r_v$, both absent in vanilla GRPO-Zero. As shown in Table 2 and Table 3, removing SFT or $r_v$ leads to clear drops, for instance, without $r_v$, accuracy on VKBench falls from 67.74 to 65.71, and removing SFT (purely GRPO-Zero) further drops it to 64.29. SFT cold start phrase establishes the structured output crucial for stable GRPO optimization, while $r_v$ directly encourages grounding in visual evidence.
>
> **Question 1:**
> As clarified in Section 3.3, the QA sources differ by task type: Human-centric tasks: The original annotations contained substantial noise and ambiguity. Therefore, all QA pairs were rewritten and revalidated using DeepSeek-R1 to ensure linguistic clarity and balanced reasoning focus. World-centric tasks: The QAs were directly adopted and reorganized from high-quality existing datasets (e.g., IntPhys2, PACS), since their original annotations are already well-validated and unambiguous.

---

### Note · Authors · 2025-11-14

I have read and agree with the venue's withdrawal policy on behalf of myself and my co-authors.